# Intraoperative Needle Tip Tracking with an Integrated Fibre-Optic Ultrasound Sensor

**DOI:** 10.3390/s22239035

**Published:** 2022-11-22

**Authors:** Christian Baker, Miguel Xochicale, Fang-Yu Lin, Sunish Mathews, Francois Joubert, Dzhoshkun I. Shakir, Richard Miles, Charles A. Mosse, Tianrui Zhao, Weidong Liang, Yada Kunpalin, Brian Dromey, Talisa Mistry, Neil J. Sebire, Edward Zhang, Sebastien Ourselin, Paul C. Beard, Anna L. David, Adrien E. Desjardins, Tom Vercauteren, Wenfeng Xia

**Affiliations:** 1School of Biomedical Engineering and Imaging Sciences, King’s College London, 4th Floor, Lambeth Wing, St Thomas’ Hospital, London SE1 7EH, UK; 2Department of Medical Physics and Biomedical Engineering, University College London, Gower Street, London WC1E 6BT, UK; 3Wellcome/EPSRC Centre for Interventional and Surgical Sciences, University College London, London W1W 7TY, UK; 4Elizabeth Garrett Anderson Institute for Women’s Health, University College London, 74 Huntley Street, London WC1E 6AU, UK; 5NIHR Great Ormond Street BRC and Institute of Child Health, University College London, 30 Guilford Street, London WC1N 1EH, UK

**Keywords:** ultrasound-guided interventions, ultrasound needle tracking, fibre-optic hydrophone

## Abstract

Ultrasound is an essential tool for guidance of many minimally-invasive surgical and interventional procedures, where accurate placement of the interventional device is critical to avoid adverse events. Needle insertion procedures for anaesthesia, fetal medicine and tumour biopsy are commonly ultrasound-guided, and misplacement of the needle may lead to complications such as nerve damage, organ injury or pregnancy loss. Clear visibility of the needle tip is therefore critical, but visibility is often precluded by tissue heterogeneities or specular reflections from the needle shaft. This paper presents the in vitro and ex vivo accuracy of a new, real-time, ultrasound needle tip tracking system for guidance of fetal interventions. A fibre-optic, Fabry-Pérot interferometer hydrophone is integrated into an intraoperative needle and used to localise the needle tip within a handheld ultrasound field. While previous, related work has been based on research ultrasound systems with bespoke transmission sequences, the new system—developed under the ISO 13485 Medical Devices quality standard—operates as an adjunct to a commercial ultrasound imaging system and therefore provides the image quality expected in the clinic, superimposing a cross-hair onto the ultrasound image at the needle tip position. Tracking accuracy was determined by translating the needle tip to 356 known positions in the ultrasound field of view in a tank of water, and by comparison to manual labelling of the the position of the needle in B-mode US images during an insertion into an ex vivo phantom. In water, the mean distance between tracked and true positions was 0.7 ± 0.4 mm with a mean repeatability of 0.3 ± 0.2 mm. In the tissue phantom, the mean distance between tracked and labelled positions was 1.1 ± 0.7 mm. Tracking performance was found to be independent of needle angle. The study demonstrates the performance and clinical compatibility of ultrasound needle tracking, an essential step towards a first-in-human study.

## 1. Introduction

Ultrasound (US) guided needle procedures have been widely applied in the clinical contexts of tumour biopsy, regional anaesthesia and fetal medicine [1]. Clear visibility of the needle tip within the US image is essential to reach the procedure target and prevent erroneous needle placement, which may lead to adverse events such as nerve damage, organ injury [2,3] or pregnancy loss [4]. In fetal medicine, 0.5% of women who undergo chorionic villus sampling or amniocentesis—two common diagnostic procedures—will miscarry, while 6% of procedures fail and need to be repeated [5].

The visibility of a needle within a B-mode, pulse-echo US image depends on its echogenicity relative to the surrounding tissue [3,6,7]. Two acoustic phenomena that contribute to needle echogenicity are diffuse scattering and specular reflection. The small diameter of the needle relative to the US wavelength results in diffuse scattering in planes perpendicular to the needle’s long axis, while its smooth surface and long length cause specular reflection in the plane parallel to its long axis. At steep insertion angles, specular reflection can cause US waves to miss the receiving US aperture, resulting in poor visibility [3]. Artefacts caused by reverberation and side lobes can also complicate the localisation of the needle tip [8], and at large depths, acoustic attenuation in tissue can deteriorate visibilty [9]. Even if the needle tip is well-visualised, the intersection of the needle shaft with the imaging plane may be incorrectly identified as the needle tip, causing misplacement, even during insertions performed by experienced operators [3].

Over the past three decades, a variety of passive methods have been proposed to improve needle visibility, including: techniques for distinguishing the needle from its surrounding tissues using image processing algorithms [10,11,12]; detection of a needle’s motion relative to the US probe with colour Doppler US [13,14,15,16]; US beam steering designed to reduce losses due to specular reflection [17]; and modification of a needle’s surface to enhance diffuse scattering, thereby reducing specular reflection [18,19,20]. Such methods have been reported to be effective to some extent, and surface-modified needles have seen success in the clinic [19,20]. However, reliable, clear and accurate localisation of the needle tip still poses a challenge, particularly in high echogenicity tissues [21,22], with steep insertion angles (greater than 45${}^{{}^{\circ}}$) and in heterogeneous tissues containing structures resembling needle shafts [23].

Active needle tracking methods have also been recently proposed, typically relying on either electromagnetic (EM) [24,25,26] or US transducers embedded within the needle shaft and tip [1,27,28,29,30,31,32]. In EM tracking, a magnetic sensor is located at the needle tip and exposed to a known EM field. The magnetic gradient detected by the sensor is then used to derive the spatial position of the sensor within the field. This is registered to the coordinates of the US probe, which is fitted with its own sensor. This method is, however, sensitive to environmental EM field disturbances—for example from surgical instruments [33]—and its resolution can exceed 3 mm [34,35]. The requirement for an external EM field generator and a tracking sensor on the US probe is also disruptive to the clinical workflow. Winsberg et al. introduced US needle tracking in 1991 with a piezoelectric element located at the needle tip; when struck by a transmission from the US probe, the electrical signal from the receiver was delayed and injected into the echo train of the US imaging system, causing a bright echo to appear in the US image at the needle tip position [36]. Later work by Langberg et al. utilised a piezoelectric transponder that returned acoustic signals to the US probe [1]. Recent advancements in fibre-optic US devices have enabled the development of similar methods utilising photoacoustic transmitters, receivers and transponders with the potential to provide sub-millimetre accuracy [32,34,37]. We previously presented an ultrasonic tracking technique based on a fibre-optic hydrophone (FOH) integrated into an intraoperative needle [28,29,30]. The system converted a temporal ultrasonic signal received at the needle tip into a spatial position which was clearly visualised as a cross-hair on the US image [28,29,30]. However, the system required a dedicated US transmission sequence for tracking that was interleaved with the imaging sequence. A later system utilised a bespoke 1.5D US imaging probe [38,39]. Both techniques required the use of a research US platform, featuring only simple B-mode US image reconstruction and therefore degraded image quality, which impeded their use in the clinic.

This paper presents a next-generation, real-time FOH tracking system capable of locating the needle tip within the imaging plane of a commercial US system (Voluson E10, GE, Zift, Austria), using only limited knowledge of the transducer geometry and transmission sequence. The system has been developed under the ISO 13485 Medical Devices quality standard [40], and is intended for application in fetal-medicine diagnostic procedures such as chorionic villus sampling and amniocentesis. Three in vitro (homogenous medium) and two ex vivo (heterogeneous medium) experiments assessing aspects of tracking accuracy, directivity and signal-to-noise ratio are presented and their results discussed. Two experiments related to safety are also presented: temperature rise at the FOH tip was measured in vitro in a water phantom, and ex-vivo placenta tissue was histologically examined after exposure to the FOH, demonstrating that no tissue damage was caused by laser-induced heating from the FOH.

## 2. Us Needle Tracking (UNT) System

### 2.1. Hardware

The US needle tracking (UNT) system is appended to a commercial US imaging system (Voluson E10, GE, Zift, Austria) with a convex US probe (C1-5-D, GE, Zift, Austria). For the current study, the imaging system was configured with an imaging depth of 300 mm and field of view of 55${}^{\circ }$, resulting in a US frame rate of 18 Hz [41]. A diagram of the components of the UNT system is shown in Figure 1a. The core component is a fibre-optic hydrophone (FOH) integrated into the stylet of a 20 gauge 150 mm long needle cannula intended for chorionic villus sampling, shown in Figure 2. The stylet and FOH are intended to be removed from the cannula after placement of the tip at the clinical target. The FOH comprises a Fabry-Pérot cavity (a polymer spacer between two reflective layers, outer diameter 150 μm) at its distal end, the optical reflectivity of which changes proportionally to incident acoustic pressure [42]. The FOH is integrated into the stylet with its sensitive element aligned to the needle tip [43], affixed at both the distal and proximal ends using medical grade epoxy. At the distal end, the epoxy both functions as an acoustic couplant and, in the event that the FOH is damaged, contains material from the FOH within the stylet. The needle with integrated FOH was assembled by OxDevice Limited (Abingdon, UK). The Fabry-Pérot cavity is interrogated with a wavelength-tunable laser (1500 nm to 1600 nm wavelength, TSL-550, Santec, Komaki, Japan), and reflected light is delivered to a photo-detector, inducing a voltage signal proportional to the time-varying reflectivity of the cavity and therefore acoustic pressure [42,44,45].

A digitizer (M2p.5961-x4, Spectrum Instrumentation GmbH, Großhansdorf, Germany) is used to acquire the voltage signal provided by the photo-detector (the “FOH signal”) with a sample rate of 10 MHz (the centre frequency of the US pulses is around 2 MHz). The US imaging system provides a “line trigger”: a voltage trigger signal synchronised to each US transmission, provided via a modified transducer cable. This voltage signal is integrated by bespoke electronics to provide a “frame trigger”—a gate signal synchronised to each US frame, i.e., each set of US transmissions used to generate a B-mode US image—as shown in Figure 1b. Both the raw and integrated trigger signals are provided to the digitizer and used to split the received signal into waveforms corresponding to each US transmission from the imaging system, and correctly group them into frames. The imaging system is also connected to a frame grabber (DVI2PCIe Duo, Epiphan Systems, Ottawa, Canada) with an HDMI cable. The frame grabber acquires B-mode ultrasound images from the imaging system, and is installed alongside the digitizer within a PC workstation. The acquired FOH signals are processed by the PC and used to superimpose a cross-hair cursor onto each acquired US image at the position of the needle tip as shown in Figure 1c.

### 2.2. Tracking Algorithm

Each acquired set of FOH waveforms is arranged into a 2D array referred to as an “FOH frame” and used to derive a needle position. Each column of the frame corresponds to a time-domain waveform received from a US transmission along a scan-line at a particular angle of the B-mode image (relative to the centre of curvature of the US probe). The signal processing pipeline and tracking algorithm are described in detail in [46] and are summarised here.

The FOH waveforms in the frame are pre-processed with a matched filter and an envelope detector to maximise the signal-to-noise ratio (SNR) and reduce jitter in the determination of time-of-flight (a later step in the tracking algorithm). The template signal for the matched filter was synthesised based on observation of the US pulses received by the FOH during tracking, as described in [46]. The SNR within the frame is then estimated and frames with a poor SNR are discarded.

The location of the needle tip is initially determined within the FOH frame space, in polar coordinates (r,θ) with an origin at the centre of curvature of the US probe and θ being relative to the vertical (axial) direction. In many cases, the position can be estimated simply by taking the (r,θ) coordinates of the maximum amplitude pixel of the frame. However, in the acoustic near-field this may lead to erroneous localisation due to side-lobes and off-axis maxima. Instead, if there is sufficient dynamic range, a −6 dB threshold is applied to the energy distribution across the columns of the frame, and then the centre-of-mass (i.e., weighted-mean) $\bar{\theta }$ of the thresholded distribution is taken as the angle of the needle tip:(1)$$\bar{\theta }=\frac{\sum_{n=1}^{N}\theta_{n}e_{n}}{\sum_{n=1}^{N}e_{n}}$$
where *N* is the number of waveforms in the frame, $\theta_{n}$ is the angle of the scan line corresponding to the *n*th waveform, and $e_{n}$ is the energy of the *n*th waveform in the thresholded frame. The threshold is required to prevent noisy scan-lines introducing a bias to the centre of mass calculation in either angular direction [46]. An estimate of the uncertainty of this measurement is made by determining the weighted standard deviation of the energy distribution, which is a measure of its angular width:(2)$$\sigma_{\theta }=\sqrt{\frac{\sum_{n=1}^{N}e_{n}{\left(\theta_{n}-\bar{\theta }\right)}^{2}}{\frac{1}{N^{\prime }}\left(N^{\prime }-1\right)\sum_{n=1}^{N}e_{n}}}$$
where $N^{\prime }$ is the number of waveforms with non-zero energy. In our case, $N^{\prime }$ is always equal to *N* because some noise is always present.

The *r* coordinate is determined from the time of arrival *t* of the acoustic pulse: r=ct+ρ where *c* is the sound-speed assumed by the clinical US system and ρ is the radius of curvature of the probe. The time of arrival is determined using the method described in [47]. According to [47], the pulse duration is 1.25δt where δt is the interval between the times when the time-integral of the square of the waveform reaches 10% ($t_{0.1}$) and 90% ($t_{0.9}$) of its final value. Therefore the start of the pulse is $t_{0.1}-0.125\delta t$. The uncertainty in this measurement is taken to be the pulse length: $\sigma_{r}=1.25c\delta t$.

### 2.3. Scan Conversion

Polar coordinates (r,θ) are converted to Cartesian coordinates (x,y) relative to the centre of curvature of the probe: x=rcos(ϕ) and y=rsin(ϕ), where $\phi =\theta -90^{\circ }$. (because θ is relative to the vertical axis). In order to plot the resulting Cartesian coordinates onto the US image, it was necessary to determine the size of the pixels of the acquired US images and the coordinates of the centre of curvature of the probe within the images. These parameters were determined automatically through the acquisition and analysis of a single US frame using knowledge of the imaging depth. The uncertainties $\sigma_{r}$ and $\sigma_{\theta }$ are propagated through the scan conversion to determine uncertainty estimates $\sigma_{x}$ and $\sigma_{y}$ for the tracked Cartesian position of the needle. This first required the construction of a covariance matrix $K_{r,s}$ from $\sigma_{r}$ and $\sigma_{\theta }$ representing the covariance in Cartesian coordinates (r,s) aligned to θ, where s⊥r:(3)$$K_{r,s}=\left[\begin{array}{cc} \sigma_{r}^{2} & 0\\ 0 & r\tan\sigma_{\theta }^{2} \end{array}\right].$$

To align $K_{r,s}$ to the *x* and *y* axes required a rotation by ϕ using a rotation matrix R:(4)$$R=\left[\begin{array}{cc} \cos\phi & -\sin\phi \\ \sin\phi & \cos\phi \end{array}\right].$$$\sigma_{x}^{2}$ and $\sigma_{y}^{2}$ can then be determined by rotating $K_{r,s}$ to form $K_{x,y}$:(5)$$\begin{array}{cc} K_{x,y} & =\left(RK_{r,s}\right)R^{T}\\ \; & =\left[\begin{array}{cc} \sigma_{x}^{2} & \sigma_{x}\sigma_{y}\\ \sigma_{y}\sigma_{x} & \sigma_{y}^{2} \end{array}\right]. \end{array}$$

### 2.4. Software

The tracking algorithm was implemented as part of a desktop application written in Python (version 3.9) and running on Microsoft Windows (version 10). The UNT application provides a graphical user interface (GUI) (shown in Figure 1c) comprising a real-time display of the acquired US images with superimposed tracking cursor. The system is capable of real-time tracking at the frame rate of 18 Hz that results from the chosen US imaging system settings. It should be noted that although the tracking cursor is rendered within this time frame (i.e., with a latency less than 55 ms), the frame grabber introduces a latency of approximately 170 ms to the video feed. The colour, size and opacity of the cross-hair tracking cursor are configurable and can be set to change in proportion to the estimated uncertainties $\sigma_{x}$ and $\sigma_{y}$. A “research” GUI mode provides real-time plots of the raw and processed FOH waveforms. The software has the ability to stream raw FOH waveforms and US images to disk. These stored data can then be viewed using the application’s “playback” mode—which mimics live acquisition—or analysed offline. The software module responsible for the acquisition of signals from the digitizer has since been released as an open source Python package [48].

## 3. Experimental Methods

Three in vitro and two ex vivo experiments assessing the performance of the UNT system were carried out. The in vitro experiments took place in a tank of water, aiming to assess the performance of the system in a homogenous medium. The first homogenous-medium experiment assessed tracking accuracy at 356 known locations in the imaging plane; the second assessed the out-of-plane tracking accuracy at a fixed lateral and elevational position; and the third assessed the dependence of FOH signal amplitude and tracking accuracy on needle insertion angle. In tissue, attenuation and heterogeneities in sound-speed and acoustic impedance were likely to aberrate the acoustic field and reduce signal amplitude, degrading tracking accuracy. The first ex vivo experiment therefore assessed tracking accuracy in a bovine tissue and water phantom designed to mimic a transabdominal fetal medicine procedure, by comparison to manual annotation of the apparent location of the needle tip in B-mode US images. The second heterogeneous-medium experiment assessed the dependence of SNR on depth in another phantom comprised solely of bovine tissue.

Two experiments relating to safety were carried out. The first was an in vitro experiment assessing the temperature rise at the tip of the needle in water. The second experiment histologically assessed ex vivo placenta samples for signs of thermal damage after insertion with a needle with integrated FOH. The placentas were from women undergoing elective Caesarean section at term who gave informed written consent (“Fetal Stem Cells” REC reference: 14/LO/0863. IRAS ID: 133888).

### 3.1. In-Plane Tracking Accuracy in a Homogenous Medium

For the homogenous-medium experiment, the US probe was placed over a water tank with dimensions 70 × 40 × 50 cm. A bespoke probe mount allowed manual adjustment of its pitch, yaw and roll as shown in Figure 3. The probe was held pointing vertically downwards with its face immersed in the room-temperature water to a minimum depth of approximately 5 mm. The base of the tank was lined with an acoustically absorbing material to prevent reflections. The needle was clamped to an arm attached to a 3-axis, 2.0 μm resolution computer controlled Cartesian motion system (1N150 linear stages, Thorlabs, Newton, NJ, USA) and positioned in the tank below the imaging probe. The needle was held with its long axis horizontal to the water surface, maximising the range of accessible positions within the US field of view. The needle was oriented with its bevel facing upwards, exposing the integrated FOH to incident ultrasound. The needle clamp and motion control system were positioned on the right hand side of the imaging probe, with the needle pointing left into the US imaging plane as shown in Figure 4.

#### 3.1.1. US Imaging System Configuration

The US imaging system was configured with an imaging depth of 300 mm and field of view of 55${}^{{}^{\circ}}$, resulting in a frame rate of 18 Hz [41]. These settings were chosen based on proprietary details of the pulse transmission sequence provided by GE. A single focal zone was set at a depth of 15 mm, ensuring that all measurements were made in the acoustic far-field. The acoustic output was set to the default value of 98%. Settings that did not affect the acoustic output of the probe were not recorded, as they had no effect on tracking performance.

#### 3.1.2. Determining True Needle Positions

Determination of the true needle positions relative to the US probe was achieved by alignment of the US probe to the needle’s motion axes. True, 2D needle positions within the imaging plane were determined in Cartesian coordinates (x,y) relative to the centre of curvature of the probe, *x* being the lateral direction and *y* being the axial (vertical) direction. This comprised first aligning the lateral axis of the motion control system to the imaging plane, and then aligning the central vertical axis of the US image to the axial motion of the control system.

The lateral motion of the needle tip was aligned to the imaging plane as follows: the US probe was oriented pointing vertically downwards using a spirit level; the needle tip was positioned along the central vertical axis of the US image, approximately 75 mm from the probe face; the elevational position of the needle was adjusted such that the brightness of the needle in the US mage was maximised; the needle was translated laterally by 75 mm; the roll of the probe was adjusted such that the brightness of the needle in the US mage was maximised. Success was confirmed by translating the needle tip laterally from one side of the imaging plane to the other and observing that its brightness in the US image did not change.

An axial alignment procedure was designed based on the iterative “near-far” method for aligning the beam axis of a single-element, axisymmetric US source to the axial motion of a hydrophone described in [49]. The cited axisymmetric alignment procedure comprises repeatedly moving the hydrophone between two distantly-spaced far-field positions nominally along the beam axis, maximising the US signal at each position by making adjustments to either the position of the hydrophone or orientation of the source. After several iterations, the axial motion of the hydrophone and the beam axis of the source will converge. This alignment procedure was adapted to allow alignment of the central vertical axis of the B-mode US image to the axial motion of the FOH. The needle was translated between 40 mm and 140 mm nominally directly below the centre of the probe face, with the distance determined from the time of arrival of the US pulse to the FOH embedded in the needle tip combined with knowledge of the delay between the trigger signal and US transmissions internal to the imaging system, and the radius of curvature of the probe; although no pulse was emitted from the centre of the probe, the scan lines either side of the central axis were very close to the centre and as such their use introduced a negligible error. At the 40 mm depth, the lateral and elevational positions of the needle were adjusted to position its tip on the central axis. At the 140 mm depth, the pitch and yaw of the probe were adjusted to coincide the central axis with the needle tip position.

At each iteration, an approximate alignment of the FOH tip to the central vertical axis could be achieved by adjusting the position of the needle or orientation of the probe until the B-mode image of the needle tip coincided with the central axis of the image, using a rigid vertical marker placed over the screen of the imaging system as a guide. However, although the focal depth of the US imaging system was adjusted at each iteration to optimise the resolution of the US image at the needle depth, US image artefacts around the needle tip made it difficult to precisely locate the needle tip using the B-mode image alone. Following this, a more precise alignment was achieved by examining the FOH signal visualised with an oscilloscope, using the signal amplitude to find the midpoint (i.e., the amplitude minimum) between the two central scan-lines of the US imaging frame.

#### 3.1.3. Data Collection

It was expected that tracking accuracy would depend on both the depth and lateral position of the needle tip within the imaging plane: the probe’s curvilinear design results in the spacing between scan-lines increasing with both depth and lateral distance from the central vertical axis of the scan plane. Tracking accuracy was therefore assessed at 356 locations within the imaging plane, equally spaced across the right-hand side plane between 40 mm and 140 mm below the probe face, forming a 5 mm resolution grid as shown in Figure 5. The fetal medicine procedure of particular interest—chorionic villus sampling—typically occurs between 11 to 16 weeks of gestation with the target located well within this depth range [50], and previous US needle tracking systems have been assessed over a similar range [30,39]. Due to physical constraints of the motion control system and hardware setup, it was not possible to laterally translate the needle across the whole field of view, but in any case tracking accuracy could be expected to be symmetrical along the central vertical axis of the probe.

At each position, 18 frames (i.e., 1 s) of FOH waveforms were acquired providing 18 measurements of the position of the needle tip. Five repeats of this experiment were carried out, with the needle and probe being removed and remounted and the above alignment procedure being repeated between each.

#### 3.1.4. Data Analysis

Two accuracy metrics were determined at each of the 356 locations: tracking error e and tracking repeatability σ. Tracking error was defined as the vector displacement between the tracked and true positions, averaged over the 18 measurements. Tracking repeatability was defined as the standard deviation of the 18 tracked positions.

It was expected that, due to the use of a convex probe, any systematic tracking error introduced by the US imaging system would likely manifest as a constant error in polar space. This would result in a spatially-varying tracking error in Cartesian space. Tracking error and repeatability were therefore assessed in both polar and Cartesian coordinates. Each true needle tip location was derived from the alignment procedure in Cartesian coordinates (x,y) relative to the centre of curvature of the probe. These were directly compared with the Cartesian, scan-converted, tracked positions. For comparison with pre-scan-conversion polar tracked positions, the Cartesian true positions were converted to polar coordinates (r,θ):(6)$$\begin{array}{cc} r & =\sqrt{x^{2}+y^{2}}\\ \theta & =\cos^{-1}\left(\frac{y}{r}\right). \end{array}$$

Tracking error e was then determined in Cartesian coordinates as
(7)$$e=\left[\begin{array}{c} \bar{\hat{x}}-x\\ \bar{\hat{y}}-y \end{array}\right]$$
and polar coordinates as
(8)$$e=\left[\begin{array}{c} \bar{\hat{r}}-r\\ \bar{\hat{\theta }}-\theta \end{array}\right]$$
where $\left(\bar{\hat{x}},\bar{\hat{y}}\right)$ and $\left(\bar{\hat{r}},\bar{\hat{\theta }}\right)$ were the average of the 18 tracked position in Cartesian and polar coordinates respectively. Repeatability σ was simply the standard deviation of the 18 individual measurements of tracked position, $\left(\hat{x},\hat{y}\right)$ or $\left(\hat{r},\hat{\theta }\right)$.

At each location, five measurements of e and σ were made—one from each of the five repeat datasets. These five measurements of error and repeatability were averaged for each location. The random uncertainty in the measurement of e was estimated by dividing the standard deviation of the repeat measurements of e by $\sqrt{N}$, where N=5 (the number of repeat measurements).

The final values of error and repeatability for each position were then averaged across the whole scanned area, and the standard-deviation taken to provide single values for the spatial variation of tracking error and repeatability respectively. The absolute value of the spatial-average tracking area in each direction was also of interest.

In summary:Mean and standard deviation at each position provide the spatially-varying tracking error and repeatability of the systemThese metrics were measured five times and averaged, with the standard deviation across the five repeats used to estimate the measurement uncertaintySpatial average and standard deviation of tracking error and repeatability were also taken to provide single values for the systems tracking performance.

### 3.2. Out-of-Plane Tracking Accuracy in a Homogenous Medium

The experimental setup and alignment for the assessment of out-of-plane tracking accuracy was identical to that of the in-plane tracking accuracy assessment described above. The needle tip was initially positioned 145 mm from the probe face along the central vertical axis of the US image, and then translated elevationally to discrete positions either side of the imaging plane, and tracking accuracy was assessed at each position. The experiment was carried out with two different US imaging focal-depths: 15 mm and 140 mm, expecting that the out-of-plane tracking performance may depend the shape of the ultrasound beam profile at the needle tip depth. An FOH frame was also acquired with the US output disabled (a “noise” dataset), for determination of signal-to-noise ratio.

#### 3.2.1. Data Collection

The needle tip was scanned along a straight line covering elevational positions from −36 mm to 36 mm in 1 mm steps, where the US imaging plane was located at an elevational position of 0 mm. At each position, a set of 18 FOH frames was acquired. This process was repeated three times for each focal depth, with the needle removed and realigned between each repeat.

#### 3.2.2. Data Analysis

At each position, the tracked positions determined from the 18 FOH frames were averaged and compared to the true in-plane position, resulting in one axial tracking displacement and one lateral tracking displacement. The standard deviations of the tracked lateral and axial positions were taken as the tracking repeatabilities in those directions.

Noise was expected to be independent of position, so common values for noise level were used in the calculation of SNR at all positions. The root-mean-square (RMS) voltage in each noise waveform of the acquired frame was determined, and then these RMS values were averaged to provide the common noise level. Signal-to-noise ratio in each measurement frame was the calculated by dividing the peak voltage in then frame by the noise level.

The means and standard deviations of the tracking displacements, repeatabilities and signal-to-noise ratios were then taken across the three repeat measurements, with the standard deviations indicating the measurement uncertainty.

### 3.3. Assessment of the Effects of Needle Insertion Angle

The angular dependence of tracking accuracy and FOH signal amplitude were assessed using the same experimental setup and imaging system configuration as the homogenous-medium tracking accuracy assessments described above. The needle mount included a miniature rotational stage that allowed the angle of the needle to be set with a precision of approximately ±5${}^{{}^{\circ}}$. For all angles, the needle shaft remained aligned to the imaging plane.

#### 3.3.1. Data Collection

A set of 18 FOH frames was collected with the needle at each of thirteen angles equally spaced between 0${}^{{}^{\circ}}$ and 60${}^{{}^{\circ}}$ relative to horizontal. For each acquisition, the needle tip was positioned directly below the US probe, 150 mm from the probe face, with the depth determined using the time of arrival of the US pulse as described above for the tracking accuracy experiment. This was the minimum depth at which the needle tip could be positioned at all angles of interest, due to physical constraints imposed by the experimental setup. The experiment was repeated three times, with the needle removed and realigned between each repeat.

#### 3.3.2. Data Analysis

The averaged of the amplitude of the 18 waveforms acquired in each measurement was taken, before and after processing with the matched filter and envelope detector. The five repeat measurements of amplitude at each angle were averaged, and the standard deviation taken to assess the measurement uncertainty.

### 3.4. Tracking Accuracy in Tissue

Ex vivo assessment of tracking accuracy took place in a phantom comprising a layer of room temperature bovine tissue held at the surface of a small tank of room temperature water, designed to simulate transabdominal fetal-medicine diagnostic procedures such as chorionic villus sampling and amniocentesis. A schematic diagram of the phantom is shown in Figure 6. The tissue layer was approximately 50 mm thick and the water below it approximately 200 mm deep. The base of the tank was lined with an acoustically absorbing material to prevent reflections. While the needle tip was in the water layer and therefore clearly visualised, tracked positions were compared to the apparent position of the needle tip within acquired B-mode images. The configuration of the US imaging system was identical to that used for the in vitro experiment.

#### 3.4.1. Data Collection

The imaging probe was fixed in a clamp with its face pressed onto the top surface of the tissue pointing vertically downwards, coupled with commercially available US gel. The UNT software was configured to record its received data to disk, and a needle insertion was made into the top surface of the tissue at an angle of 55${}^{{}^{\circ}}$ relative to horizontal, down to a depth of approximately 150 mm, aiming to keep the needle shaft within the imaging plane throughout. The total recording time was 46 s, during which the needle was inserted and withdrawn.

#### 3.4.2. Determining Reference Needle Positions

The UNT software includes a feature that allows recorded datasets to be stepped through frame by frame, and a manually-labelled position to be annotated on each frame by clicking on the correct position in the US image. These annotated positions are then stored to disk alongside their associated frames. This feature was used to generate reference needle positions for the recorded dataset. Only frames for which the needle tip was in the water layer of the phantom, where the tip was clearly visualised, were annotated (793 out of a total of 832 frames). During the homogenous-medium experiment, it was noticed that the visually-apparent location of the needle tip in the US image was often displaced along the needle shaft from the location of the tip of the FOH (as identified from examination of the FOH signal). This apparent extension of the needle length in the US image was likely caused by diffuse reflection at the needle tip and the point spread function of the imaging system. This offset was quantified by manually labelling four datasets from the homogenous-medium experiment (with the needle tip positioned on the central vertical axis) and determining the displacement along the needle shaft between the true positions and the manual labelled positions. For the tissue dataset, manually-labelled positions were corrected for this effect by subtraction of this displacement, after rotating it by 55${}^{{}^{\circ}}$ (the difference between the angles of the needle in the homogenous-medium and tissue experiments).

#### 3.4.3. Data Analysis

The reference and tracked positions in each frame were compared as described for the in vitro experiments, although no assessment of repeatability was possible as the needle tip moved each frame. The average distance between the tracked and reference positions across the whole dataset was calculated.

### 3.5. Assessment of Signal-to-Noise Ratio and Repeatability in Tissue

Signal-to-noise ratio (SNR) and tracking repeatability were assessed in a 150 mm deep ex vivo phantom comprising four vertically stacked, approximately 4 cm thick pieces of bovine tissue. SNR was measured with the needle tip manually inserted to 15 locations along the central vertical axis of the US image. The US imaging system configuration was identical to that used for the ex vivo accuracy experiment described above.

#### 3.5.1. Data Collection

For all 15 depths, the needle was manually inserted from the side of the phantom with the shaft horizontal. This was done to ensure the angle of incidence of US at the needle tip did not contribute to any difference in SNR between positions. Once the needle had been inserted, the position of the US imaging probe was adjusted to maximise the brightness of the needle tip in the US image, ensuring it was in imaging plane. At each depth, 18 frames of FOH waveforms were acquired with the US output of the imaging system enabled (“signal” datasets), and another 18 frames were acquired with the US output disabled (“noise” datasets).

#### 3.5.2. Data Analysis

SNR was calculated for both raw and processed (matched-filtered and envelope-detected) FOH waveforms. Noise was independent of position, so common values for raw and processed noise level were used in the calculation of SNR at all positions. The root-mean-square (RMS) voltage in each noise waveform was determined, and then these RMS values were averaged to provide the common noise level. The standard deviation of RMS voltage across the waveforms was also take to ensure no significant variation in noise level.

SNR was only determined for the maximum amplitude FOH waveform in each “signal” frame. The SNR was calculated by dividing the peak waveform voltage by the corresponding (i.e., raw or processed) noise level. The SNR values determined at each position were then averaged, resulting in a raw and processed SNR value for each position.

Manually labelling of the needle tip positions was challenging because US attenuation and sound-speed heterogeneities within the tissue resulted in poor visualisation of the needle tip. As such, tracking error was not determined. Tracking repeatability, however, was determined at each position as described for the in vitro experiment and compared to the signal-to-noise ratio.

The validity of the tracking uncertainties estimated by the UNT system during this experiment was assessed by comparison to the determined repeatabilities. The absolute values of the two metrics were compared, and the Pearson correlation coefficient was calculated.

### 3.6. Measurement of Laser-Induced Heating

During normal operation, the infrared beam of the interrogation laser remains enclosed within the FOH, reflected by the Fabry-Pérot cavity at the tip of the fibre. The temperature rise of the FOH tip was measured using a thermocouple in water to assess the potential for laser-induced heating of the cavity to cause tissue damage in vivo.

Water was used as a medium, having a similar infrared absorption spectrum to tissue [51]. If the Fabry-Pérot tip was damaged, exposing the fibre tip, then light would be emitted from the fibre, being quickly absorbed by the surrounding tissue and causing heating. Temperature rise in water was therefore also measured at the tip of a bare optical fibre—identical to that used in the manufacture of the FOH—connected to the interrogation laser. Resulting temperature rises were compared to the limit of 6 ${}^{{}^{\circ}}$C set for invasive diagnostic ultrasound devices by the relevant IEC safety standard [52].

In each case, the fibre or FOH tip was submersed in a beaker of approximately 200 mL of water, positioned approximately 1 mm from the water surface. A wire thermocouple was placed close to (<1 mm from) the fibre or FOH tip and connected to a temperature logging system (TC-01, National Instruments, Austin, TX, USA) with a resolution of 0.1 ${}^{{}^{\circ}}$C. The interrogation laser was enabled at its maximum possible power of 25 mW; this power setting—several times greater than that used during the tracking performance assessments—was chosen so that higher laser powers could be used to boost the signal-to-noise ratio (and therefore tracking accuracy) in future if required, without needing to repeat the experiment. The temperature rise (relative to the temperature at switch-on) was recorded with a rate of 1 Hz for a period of 30 min. The maximum increase in temperature observed during the 30 min period was then logged. This procedure was repeated three times for both the FOH and bare fibre.

### 3.7. Heating of Ex Vivo Placenta Tissue

To ensure that laser-induced heating would not cause tissue damage in vivo, 25 mm × 25 mm × 10 mm excised samples of fresh placenta were exposed and then histologically examined for signs of damage to the chorionic villi. Six samples were exposed: three to heating from an FOH and three to the light emitted from a bare optical fibre. Exposures were carried out by inserting an amniocentesis needle with integrated FOH or bare fibre a few millimetres into each sample as shown in Figure 7. With the needle inserted, the interrogation laser was enabled at its maximum power of 25 mW for 10 min. To assist with histological examination, the location of the insertion was marked with a small spot of white paint. The needle was then extracted, and the exposed samples were placed into tissue cassettes and submersed in 4% *w*/*v* formaldehyde.

For histological examination, samples were processed, cut (4 μm slice thickness) and stained with Haemotoxylin and Eosin. The marked location of the needle insertion was used as a guide for the cutting of each sample. The site of the needle tip was then identified from the artefactual architecture change in the villi caused by the insertion of the needle, and the surrounding tissue was inspected for thermal damage such as coagulative necrosis and heat diathermy.

## 4. Results

### 4.1. In-Plane Tracking Accuracy in a Homogenous Medium

The average repeatability of the 18 tracked positions measured at each location are plotted in Figure 8. The spatial mean tracking repeatability magnitude was 0.28 ± 0.17 mm, the components of which are shown in Table 1. The minimum and maximum magnitudes of tracking repeatability measured across all scanned positions were 0.002 mm and 1.14 mm respectively.

The average vector displacements between the true and tracked positions measured during the in vitro experiment are plotted in Figure 9a. The estimated measurement uncertainties in the determination of this spatially-variant tracking error are plotted in Figure 9b. The minimum and maximum magnitudes of tracking error measured across all scanned positions were 0.01 mm and 2.28 mm respectively. The estimated random measurement uncertainty magnitude varied across the scanned positions from 0.28 mm to 1.49 mm, with a spatial average magnitude of 0.58 mm and spatial standard deviation of 0.24 mm. The magnitude of measurement uncertainty was less than 1 mm for 95% of positions. The spatial mean tracking error magnitude was 0.67 ± 0.41 mm, the components of which are shown in Table 1.

### 4.2. Out-of-Plane Tracking Accuracy in a Homogenous Medium

The results of the out-of-plane accuracy experiment are shown in Figure 10. With the imaging system focal depth set close to the depth of the needle tip, signal-to-noise ratio remained above 30 dB out to 30 mm either side of the imaging plane. For this configuration, the UNT system was able to maintain tracking across all elevational positions tested, with repeatabilities remaining close to ±0.5 mm in both the lateral and axial directions. Tracking displacement increased as the needle tip moved away from the imaging plane, to a maximum in the axial direction of approximately 2 mm and in the lateral direction approximately 4 mm.

With the imaging system focal depth set close to the probe face and therefore the needle tip in the acoustic far field, signal-to-noise ratio was significantly reduced, falling to 30 dB by 15 mm either side of the imaging plane. Axial tracking repeatability remained close to ±0.5 mm out to 20 mm either side of the imaging plane, and then began to increase to beyond ±4 mm. Axial displacement remained around ±4 mm out to 30 mm from the imaging plane. However, lateral repeatability and displacement indicated good tracking performance only out to approximately 15 mm either side of the imaging plane. Further than that, lateral tracking appears to fail, with both repeatability and displacement suddenly increasing.

For visibility, error bars are not shown in Figure 10. Standard deviations across the repeat measurements indicated that the alignment procedure was repeatable and therefore measurement uncertainty was low, with high standard deviation (>1 mm) only at positions with poor tracking repeatability.

### 4.3. Assessment of the Effects of Needle Insertion Angle

The average amplitudes of the waveforms acquired during the insertion angle experiment are plotted in Figure 11a. The amplitude of the raw FOH signal was found to be dependent on insertion angle, increasing by 67% between 0${}^{{}^{\circ}}$ and 40${}^{{}^{\circ}}$. However, after processing with the matched filter and envelope detector, angular dependence was much reduced, with signal amplitudes varying with angle by approximately ±10% between 0${}^{{}^{\circ}}$ and 60${}^{{}^{\circ}}$.

Two of the waveforms acquired during the needle angle experiment are plotted before, part-way through and after processing in Figure 11b, in both the time and frequency domains. The first waveform was acquired with the needle at 0${}^{{}^{\circ}}$ and the second with the needle at 40${}^{{}^{\circ}}$. The plots demonstrate that the matched-filter is responsible for creating a more uniform directional response of the needle-embedded FOH by removal of a low frequency component only present when the needle is angled.

### 4.4. Tracking Accuracy in Tissue

Figure 12c shows a representative B-mode US image from the homogenous-medium experiment used to determine any systematic error in the manually-labelled positions. The average displacement across the four datasets assessed was 3.5 mm in the direction of the needle shaft, with a standard deviation of 0.25 mm. The manual labelled positions in the tissue experiment dataset were corrected by this amount.

The trajectories of the needle tip during the heterogeneous-medium accuracy experiment, as measured using the UNT system and manual-labelling of the US images, are shown in Figure 12. The layer of porcine tissue at the top of the phantom is clearly visible in the figure. The average distance between tracked and labelled positions was 1.06 mm with a standard deviation of 0.68 mm.

### 4.5. Assessment of Signal-to-Noise Ratio and Repeatability in Tissue

The average SNR measured at each of the six depths within the bovine tissue phantom are plotted in Figure 13. The SNR was greater than 30 dB for all depths tested. The magnitude of tracking repeatability at each of the depths tested is also plotted, and ranged from 0.2 mm to 0.5 mm. The SNRs and repeatabilities shown at depth here are considered to be worst-case because of the short (20 mm) US focal depth used.

The uncertainties estimated by the UNT system software were well correlated with the measured repeatabilities (Pearson coefficient 0.72), but their values were significantly higher, ranging from range ±2.3 mm to ±5.5 mm.

### 4.6. Measurement of Laser-Induced Heating

The average maximum temperature rise across the three repeat measurements made was 0.1 ${}^{{}^{\circ}}$C for both the FOH and bare fibre, well below the 6 ${}^{{}^{\circ}}$C IEC limit [52]. In both cases, the standard deviation across the three repeat measurements was 0.1 ${}^{{}^{\circ}}$C.

### 4.7. Heating of Ex Vivo Placenta Tissue

A representative histological section of exposed villi is shown in Figure 14. Histological examination of the placenta sections showed no morphological features indicating coagulative necrosis or heat diathermy that would indicate damage due to heating. Only physical separation of the villi due to the needle insertion was apparent in any sections. There was no subjective morphological difference in histological appearance between the two groups of samples.

## 5. Discussion

We have presented the first performance assessment of a fibre-optic US needle tracking system compatible with a commercial US imaging system and which is therefore directly translatable to the clinic. The technology has been developed under the ISO 13485 Medical Devices quality standard [40] to assist minimally-invasive interventional procedures in fetal medicine such as chorionic villus sampling and amniocentesis, where automated tracking of the needle tip could reduce the frequency of adverse events such as miscarriage. During operation, the system sits alongside the US imaging device, connected to it with an HDMI cable and BNC trigger cable. The reusable components comprise a PC workstation, signal acquisition electronics and a laser source assembled into a Karl Storz (Tuttlingen, Germany) medical stack trolley. Custom single-use needles with integrated FOHs are provided sterilised and individually packaged. When in use, these are connected to the system with a thin fibre-optic cable.

The FOH embedded in the needle tip provides several advantages over alternative active US tracking technologies such as those based on an integrated piezoelectric element [31]. It provides a small active area and therefore a broad directivity [53]—reducing the likelihood that needle angle will affect tracking variance—without compromising sensitivity; when using piezoelectric elements, a smaller element broadens directivity while reducing sensitivity. The FOH also has a broadband frequency response with high sensitivity up to at least 50 MHz [53], allowing a single fibre to be used for clinical procedures of all typical US centre frequencies; this is in contrast to the narrow response of many piezoelectric elements. Delivery of signals from the sensor tip to the rest of the tracking system is trivial, as the optical signal can be examined at the end of the narrow (150 μm outer diameter) fibre which fits readily within the intraoperative needle; piezoelectric systems must integrate electrical connections into a bespoke needle, and utilise an element small enough to fit into or onto the needle tip. While the single-use component of the system—the FOH—is extremely low cost compared to piezoelectric transducers, one disadvantage is the cost of the wavelength-tunable interrogation laser, which is much greater than the electronics required to operate a piezoelectric element.

Previous FOH-based US tracking systems have relied on bespoke US transmission sequences dedicated to tracking [28,30] and/or modified US imaging probes [38,39]. Our next-generation system is capable of determining the needle position using only the US imaging transmissions, with the only modification to the US imaging system being the provision of a trigger signal. This overcomes a major hurdle in clinical translation: clinicians are able to use a familiar, commercial US imaging system featuring controls and image quality with which they are comfortable, rather than a research US system with unfamiliar controls and substandard image quality. While the UNT system was developed to operate specifically with a GE Voluson E10 C1-5-D convex US probe, the software was designed to allow the system to be used with other systems. The properties of the US imaging system upon which tracking is dependent can be manually changed in the UNT software using a configuration file—these include the imaging depth, scan angle (or width for a linear array probe), the number of scan lines per frame, the pulse transmission sequence, the radius of curvature of the probe, the coordinates of the centre of the probe face within the acquired US images, and the physical size of the pixels of the acquired US images. Many of these properties can be automatically obtained from analysis of a single high-gain US frame, as previously described, while others require proprietary knowledge of the US imaging system. Tracking is derived from the pulse transmissions intended for imaging, and therefore properties that affect US image resolution would also be expected to affect tracking performance, for example lateral scan resolution and focal depth. A limitation of the UNT system is that it is not currently possible to adjust (or automatically detect) the US imaging configuration during a procedure: the system must be restarted in order for changes to the configuration file to take affect, and while it may be possible to determine some US imaging properties in real time from the acquired US images or changes to the trigger signal, this has not yet been explored. US frame rates much higher than the 18 Hz for which the system was developed may also introduce tracking lag, as there would be less time for acquiring and processing each frame of data.

The UNT system’s positioning as an adjunct to existing medical devices—i.e., a US imaging system and needle—simplifies the regulatory processes required for clinical translation. The only change to the clinical procedure is the use of a needle with integrated FOH. A risk assessment carried out as part of the system’s development under the ISO quality management standard [40] highlighted two areas of risk relating to this: the risk of tissue damage due to heating from the FOH; and the risk of the presence of the optical fibre trailing from the proximal end of the needle impeding the clinicians ability to manoeuvre the needle. The results of the temperature rise and ex vivo placenta experiments demonstrate that the FOH, even without its reflective tip in place, is not capable of causing thermal damage to tissue through laser-inducer heating. Formative usability tests with the system have indicated that the trailing optical fibre does not impede the motion of the needle. A further usability study involving a larger group of clinicians is planned.

In the homogenous medium, the system was found to have a spatial-average tracking error magnitude of 0.67 mm with a spatial-average measurement uncertainty of ±0.58 mm. Ninety-five percent of the scanned spatial positions had a tracking error lower than 1.5 mm, and 95% of measurements had an uncertainty less than ±1 mm. The tracking accuracy of the previous FOH system, which utilised a bespoke 1.5D tracking array and pulse transmission sequence, was only assessed down to a depth (i.e., distance from the US probe face) of 60 mm [39], and a maximum absolute tracking error component (in any particular Cartesian direction) of 0.38 mm was reported. Restricted to this depth range, the accuracy of the current system is comparable: the lateral direction has the highest spatial-average absolute tracking error of 0.34 mm, with 95% of locations having an absolute error less than 0.9 mm. Spatial-average tracking error magnitude was 0.46 mm, with 95% of locations having an error less than 1 mm. Given the estimated measurement uncertainties, it is likely that the two systems have a similar tracking accuracy.

As expected, tracking performance was best in the radial direction; radial position was derived from time-of-flight and therefore benefited from the high sample rate and short US pulse length. Error and repeatability were worst in the tangential direction (the direction orthogonal to the radial direction), which was derived from the angle of the needle tip around the centre of curvature of the probe face and therefore adversely affected by the relatively poor spatial resolution of the radial A-lines of the US imaging system. This resulted in a poorer lateral accuracy than axial accuracy, on average. The tangential (and therefore lateral) accuracy worsened with depth as the gap between adjacent A-lines increased.

The results of the out-of-plane accuracy experiments demonstrate that, in water, signal-to-noise ratio remains better than 20 dB up to 30 mm either side of the imaging plane, even with the needle tip positioned far from the focal depth. The reduced out-of-plane tracking performance with the needle tip in the far-field is therefore likely due to the broader ultrasound beam profile, rather than loss of signal amplitude. This is evidenced by the worse lateral than axial tracking performance far from the imaging plane: the lateral ultrasound beam profile widens at elevational positions further from the imaging plane, increasing uncertainty in the determination of the centre-of-mass of the profile and therefore the lateral position of the needle tip. Axial tracking, which depends on time-of-flight measurement, is less affected. The results show that at the depth tested the UNT system is able to track the needle position within positions at least 15 mm either side of the imaging plane, and up to 30 mm either side with the needle tip in focus. With the needle tip in focus, out-of-plane signal-to-noise ratios were high enough that even with the additional attenuation introduced in tissue (shown in Figure 13), out-of-plane tracking performance would likely be similar in vivo to that observed in water.

During fetal medicine procedures, the clinician aims to keep the needle visualised within the imaging plane at all times by adjusting the position and orientation of the US probe. Knowledge of the distance of the needle tip from the imaging plane may assist with this process. In anaesthetics, it is common for the needle to be inserted orthogonally to the imaging plane [30], and therefore out-of-plane tracking is desirable. The high SNR provided by the UNT system combined with the US slice thickness enables the needle tip to continue to be tracked far outside the imaging plane. The software has the capability to visualise the tracking uncertainty or signal-to-noise ratio by changing the size, colour or opacity of the tracking cross-hair—this change in appearance of the cross-hair can be used to determine when the needle tip is in plane. Feedback from clinicians during usability studies has suggested that this method is useful for insertions into a tissue phantom. It was, however, challenging to configure the visualisation such that it was useful for indicating the distance of the needle tip from the imaging plane, and the resulting configuration was specific to that particular phantom. The method could not provide a quantitative elevational position of the needle tip, or even inform the user at which side of the imaging plane the tip was located. For these reasons, further work is required to improve the out-of-plane tracking provided by the UNT system. It is likely that the relative amplitudes of the signals received by the FOH from each aperture of the US probe encode its elevational position, as has been shown in similar work [54], and some preliminary work has started looking at using machine learning techniques to decode this, as has been demonstrated in the literature [55,56,57]. In [54], a photoacoustic beacon, rather than FOH, is embedded in the surgical instrument, and transmissions from it are received by the imaging probe and used to determine its 3D position. The current system could potentially be adapted to track in 3D using this principal. The work reported in [54] also makes use of a Kalman filter for smoothing tracked positions. The UNT software includes the option to enable a constant-velocity or constant-acceleration Kalman filter for reducing tracking jitter; however, usability tests determined that the apparent jitter in the position of the tracking cursor was preferable to the time-lag introduced by the Kalman filter.

In the heterogeneous tissue-and-water phantom, the average distance between tracked and true positions was 1.06 mm, nominally in the direction of the needle insertion. No such bias was seen in the homogenous-medium experiment. In the tissue-and-water phantom experiment, both the tracking system and the imaging system were operating under the same (incorrect) assumption of a homogenous sound-speed of 1540 m s${}^{-1}$, and therefore any displacement due to sound-speed heterogeneity affects the image and tracked positions equally and do not cause an offset between the two. This suggests that the displacement is likely a residual difference between the apparent position of the needle tip in the US image and the true position of the tip of the embedded FOH; in contrast, the homogenous experiment demonstrated better tracking performance because the ground truths were derived from FOH signals rather than the US image of the needle tip. Previous work involving manual labelling of the position of a needle tip in water estimated the accuracy of manual labelling to be approximately ±0.5 mm [46], so it is likely that human error is also a significant contributing factor. For our experiment, the needle tip was particularly difficult to locate precisely due to the short focal-depth used.

Tracking repeatability was excellent in water (spatial mean 0.28 mm) where there was little acoustic attenuation and therefore SNR greater than 60 dB. The results of measurements of SNR in the bovine tissue phantom suggest that in vitro signal to noise ratios could drop to 30 dB, resulting in worsened tracking repeatability of approximately 0.5 mm. The observed reduction in SNR is equivalent to an attenuation of approximately 1 dB cm${}^{-1}$ MHz${}^{-1}$. This is significantly higher than literature values of acoustic attenuation in soft tissues [58] and therefore suggests that the shape of the US field provided a significant contribution to the reduction in SNR. In the clinic, a clinician would be expected adjust the US focal depth to optimise the image of the needle, improving the SNR. However, the observed worst-case repeatability of approximately 0.5 mm is already smaller than the diameter of the needle and therefore should not cause issues in the clinic.

The real-time estimates of tracking uncertainty provided by the UNT system software were well correlated with but much higher than the measured tracking repeatability. While true tracking uncertainty of the system will be higher than the repeatability, including systematic as well as random contributions, the accuracy results presented here indicate that the uncertainty likely remains well below the software’s estimates, which reach up to ±5.5 mm. The uncertainty estimates are currently used to create a visual (but qualitative) indicator of tracking uncertainty. The ability to visualise a quantitative estimate of tracking uncertainty during operation would improve clinical user experience, and will be the subject of future work.

The 170 ms latency introduced to the video feed by the frame grabber has a noticeable effect, particularly when the needle changes direction. For a new UNT system currently under development, latency has been reduced to 70 ms through the use of an alternative frame grabber (ProCapture HDMI, Nanjing Magewell Electronics Co., Nanjing, China). This amount of latency is close to the threshold of human detectability [59]. The software library written to enable the use of the device within a real-time Python application has been released as an open source Python package [60].

The angle of the needle relative to the probe face was found have a significant effect on the amplitude of the raw FOH signal: signal amplitude was approximately 67% higher with a needle insertion angle of 40${}^{{}^{\circ}}$ than 0${}^{{}^{\circ}}$. The shape of the directional response seen was different to what would be expected from a bare FOH [53], presumably due to acoustic interaction with the needle tip. The amplitude of processed FOH waveforms did not have a significant dependence on insertion angle. Inspection of the FOH waveforms and frequency spectra at difference stages of processing (Figure 11b) demonstrated that the additional amplitude at 40${}^{{}^{\circ}}$ was provided by frequency components outside of the bandwidth of the matched filter, causing the matched filter to equalise the directional response of the embedded FOH. It can be concluded that changing insertion angle does not adversely affect tracking performance.

## 6. Conclusions

We have developed a system capable of automatically tracking the location of an intraoperative needle during US-guided procedures using a needle-integrated fibre-optic hydrophone. The tracking accuracy was found to be smaller than the needle diameter both in water and a phantom designed to simulate fetal-medicine diagnostic procedures such as chorionic villus sampling and amniocentesis, and independent of needle angle. The system was developed under the ISO 13485 quality standard [40] and designed as an adjunct to a clinical US imaging system, providing the image quality and usability that is expected by clinicians. It is therefore directly translatable to the clinic. Future work will investigate the efficacy of the technique in an animal model, leading towards a first in-human study. 

## Figures and Tables

**Figure 1 sensors-22-09035-f001:**
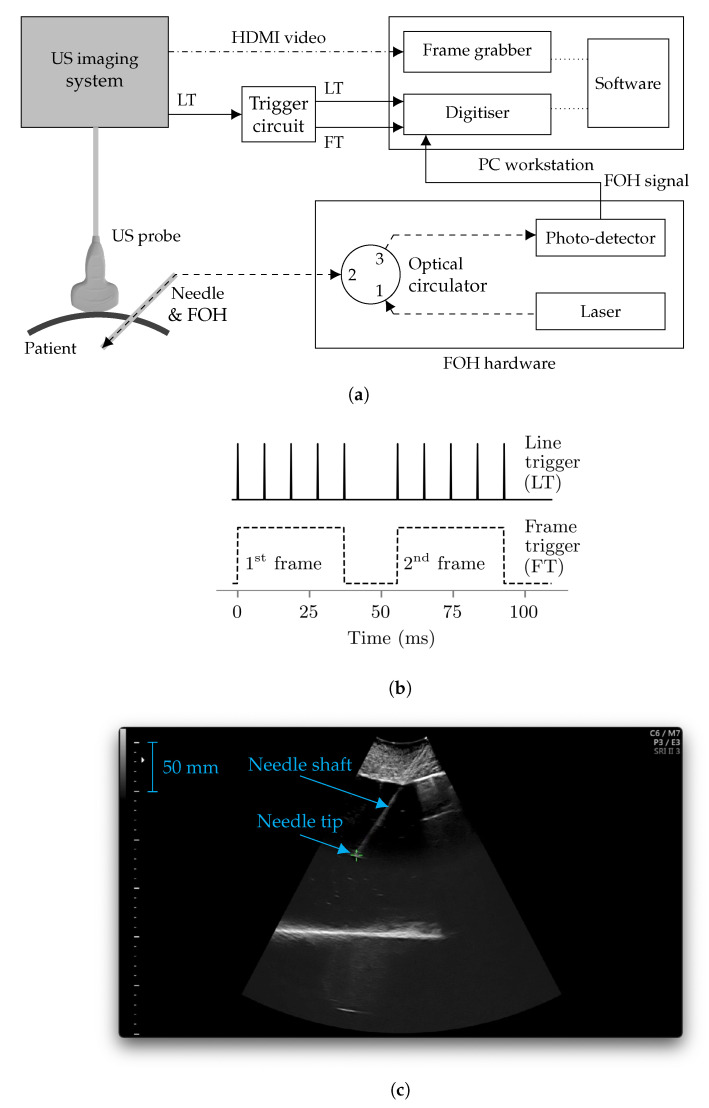
(**a**) Schematic diagram of the UNT system with the US imaging system. Solid lines represent analogue voltage signal paths, dashed lines indicate optical signal paths and the dash-dot line represents an HDMI cable. The dotted lines internal to the PC workstation represent PCIe interfaces. “LT” and “FT” respectively represent “line trigger” and “frame trigger”. Plots of the line trigger and frame trigger signals are shown in (**b**), where for visibility, there are only 5 US transmissions per frame. (**c**) An annotated screenshot of the real-time display provided to the clinician during a procedure, which indicates the tracked position of the needle tip with a green cross-hair. Blue text and arrows are additional annotations not featured in the software interface.

**Figure 2 sensors-22-09035-f002:**
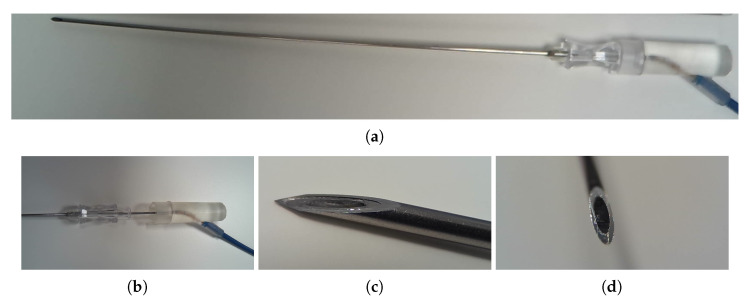
Photographs of the needle with integrated FOH: (**a**) needle assembly, with the stylet fully inserted into the cannula; (**b**) stylet partially inserted into the cannula; (**c**) the tip of the needle, with the stylet fully inserted; (**d**) the tip of the stylet.

**Figure 3 sensors-22-09035-f003:**
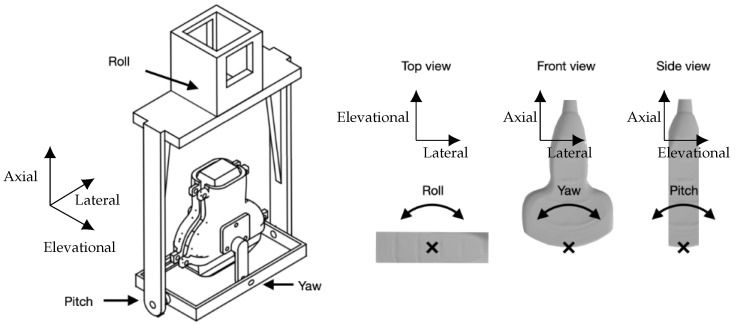
Drawing of the bespoke US probe mount allowing manual adjustment of the pitch, yaw and roll of the US probe.

**Figure 4 sensors-22-09035-f004:**
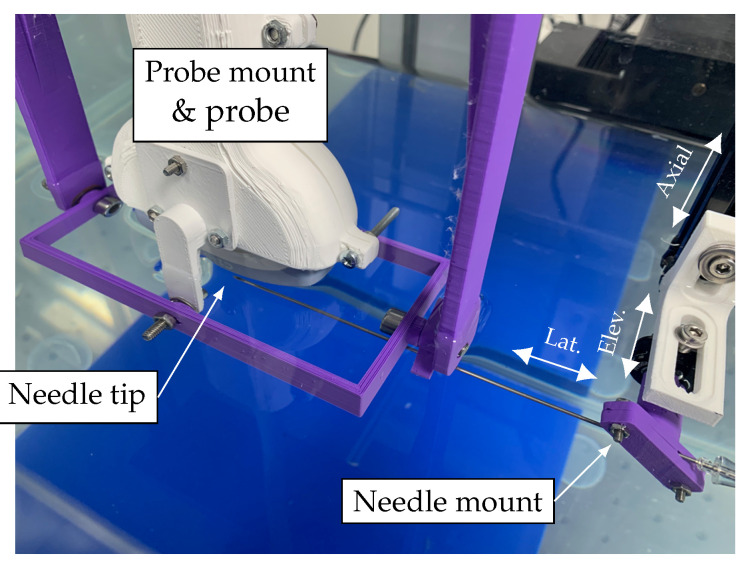
Photograph of the in vitro experimental setup showing the water tank containing the imaging probe and needle held in their mounts. The lateral, axial and elevational motion of the needle mount are indicated by white arrows.

**Figure 5 sensors-22-09035-f005:**
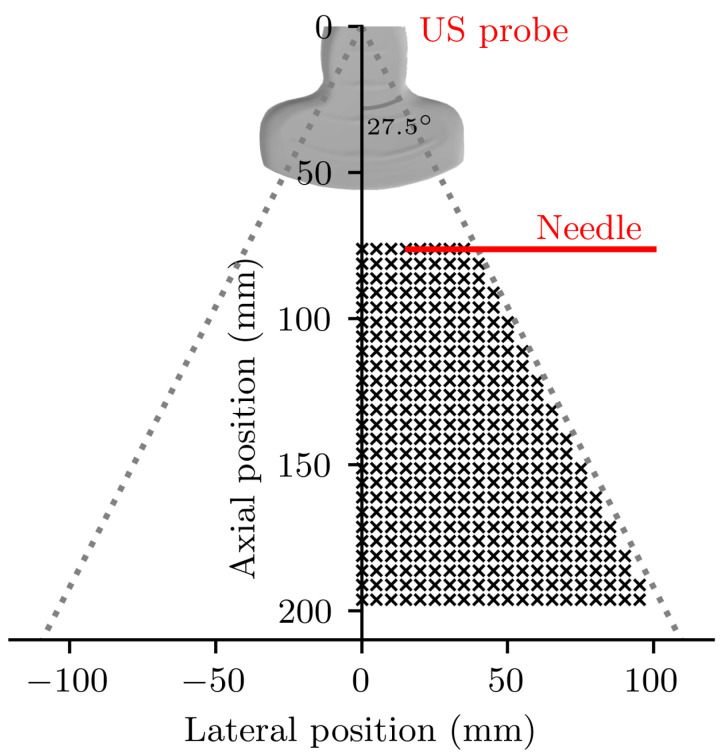
Diagram of the in-plane, homogenous medium in vitro accuracy experiment. Dotted lines indicate the extent of the field of view of the US imaging probe. Black crosses identify the locations at which tracking accuracy was assessed.

**Figure 6 sensors-22-09035-f006:**
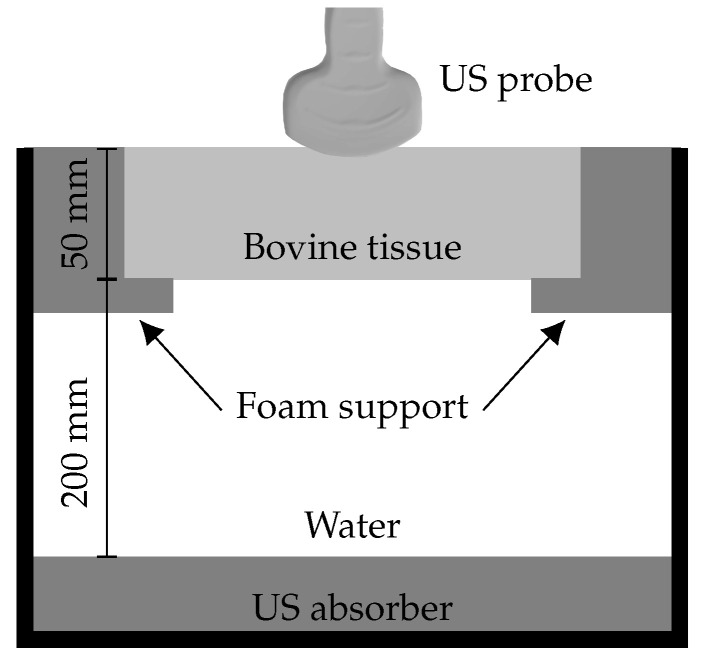
Cross-sectional diagram of the phantom used for determining tracking accuracy in tissue.

**Figure 7 sensors-22-09035-f007:**
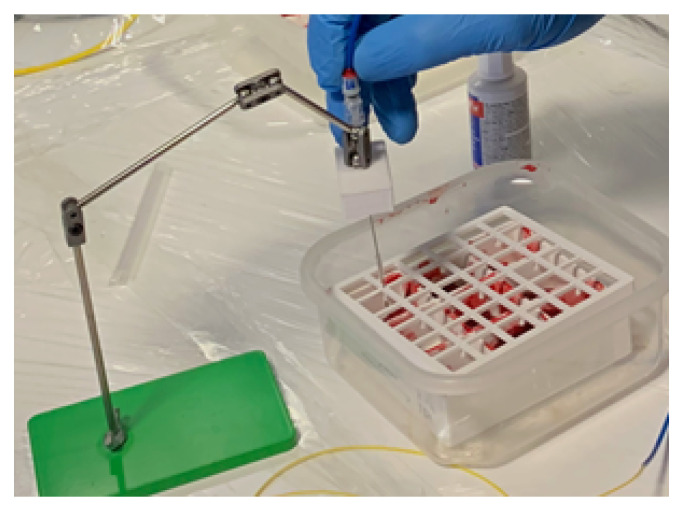
Amniocentesis needle inserted into an ex vivo placenta sample during the placenta heating experiment. The six samples were located inside a white plastic container. The container had a covering designed to support the needle during exposure.

**Figure 8 sensors-22-09035-f008:**
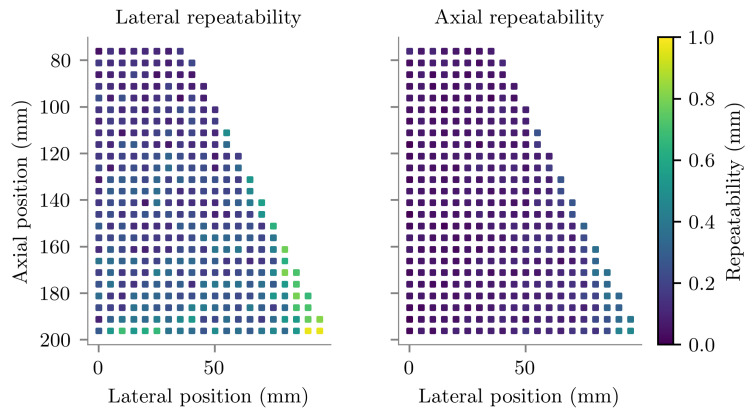
Standard deviation of the tracked lateral (**left**) and axial (**right**) locations measured in vitro, plotted at their corresponding true Cartesian positions relative to the centre of curvature of the probe.

**Figure 9 sensors-22-09035-f009:**
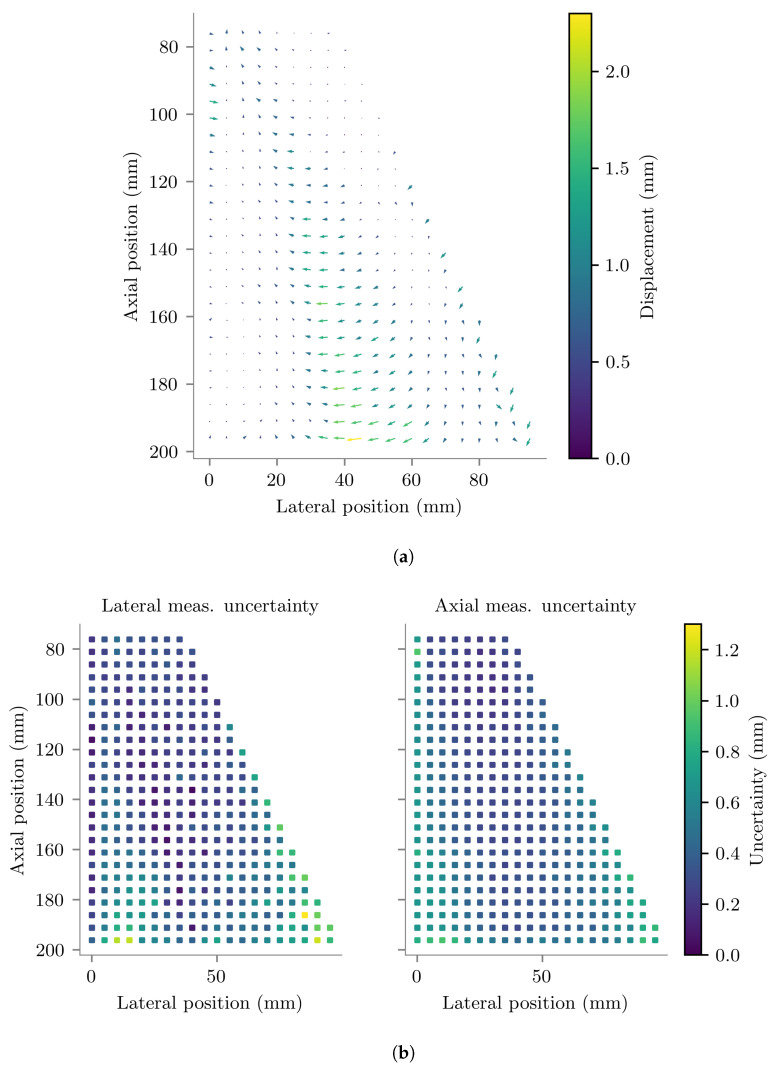
(**a**) Average vector displacement between tracked and true positions measured in water. Each arrow has its base at a true position and points in the direction of the displacement. For visibility, the arrows have been drawn with lengths equal to twice the displacement magnitude. True displacement magnitudes are indicated by the colour bar. (**b**) Random uncertainty of the measurements plotted in (**a**). Lateral and axial positions are relative to the centre of curvature of the US imaging probe.

**Figure 10 sensors-22-09035-f010:**
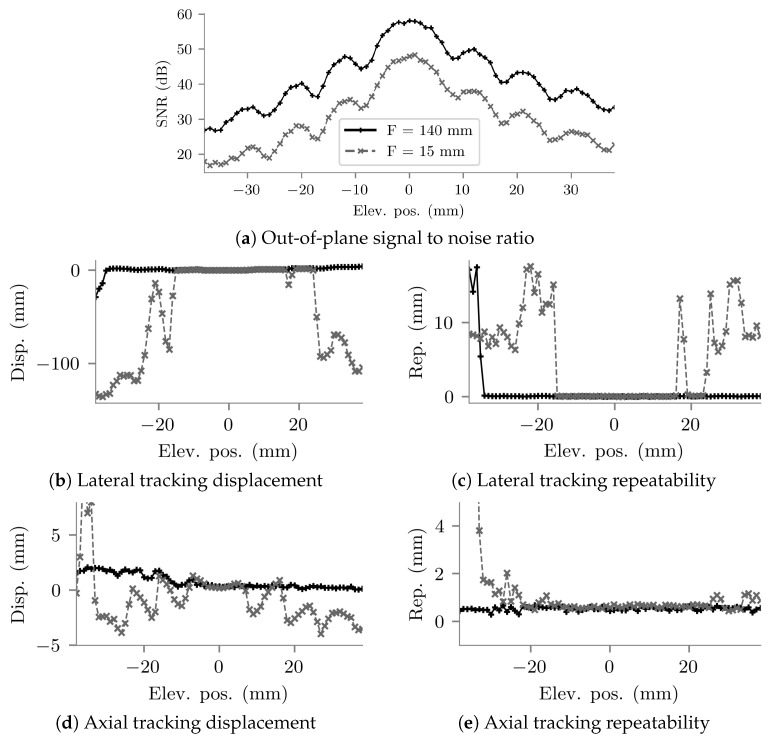
Results of the out-of-plane tracking accuracy assessment. The legend shown in (**a**) applies to all five plots. Measurements were made with two US imaging system focal depths: 140 mm and 15 mm. The needle tip was located 145 mm from the centre of the probe face in the axial direction. (**a**) Out of plane signal-to-noise ratio. (**b**) Difference between tracked and true lateral positions. (**c**) Standard deviations of the repeat measurements of lateral position. (**d**) Difference between tracked and true axial positions. (**e**) Standard deviations of the repeat measurements of axial position.

**Figure 11 sensors-22-09035-f011:**
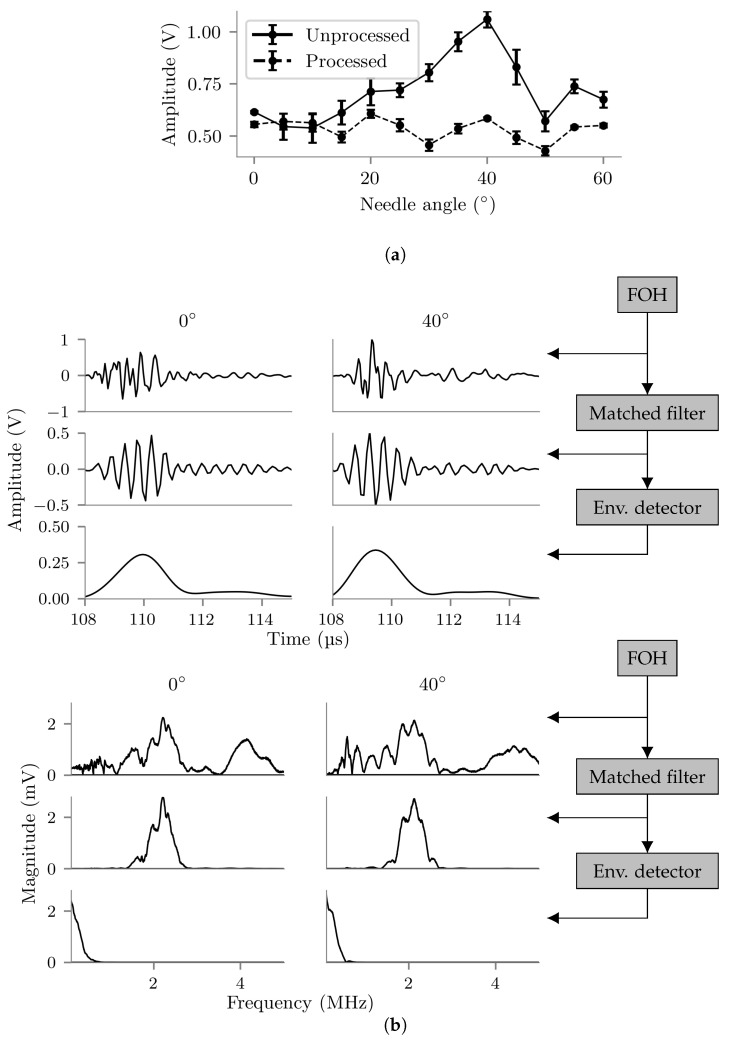
(**a**) Voltage amplitudes of unprocessed and processed FOH waveforms acquired with the needle at thirteen different insertion angles. Error bars show the standard deviation across three true-repeat measurements. (**b**) FOH waveforms (**top** figure) and frequency spectra (**bottom** figure) before processing (**top** rows), after matched filtering (**middle** rows) and after full processing (**bottom** rows) acquired with needle insertion angles of 0${}^{{}^{\circ}}$ (**left** column) and 40${}^{{}^{\circ}}$ (**right** column). Measurements were made with the needle tip located on the central vertical axis of the US image, approximately 150 mm from the imaging probe face.

**Figure 12 sensors-22-09035-f012:**
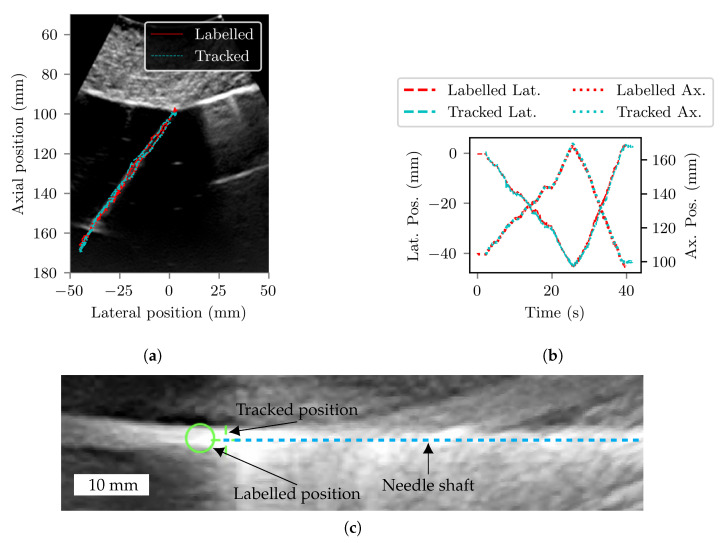
(**a**) Trajectory of the needle during the homogenous medium accuracy assessment according to manual labelling (solid red) and tracked positions generated by the UNT system (dotted blue), plotted on top of a representative B-mode US image acquired during the insertion. (**b**) Manually labelled (red )and tracked (blue) lateral and axial positions of the needle tip are plotted against time. Axial positions are relative to the centre of curvature of the US probe. A screen recording of the UNT software interface made during this insertion is provided as Appendix A. (**c**) Cropped US image of the needle during the water experiment showing the difference between tracked (in this case, approximately equal to true in) and labelled positions, and therefore the difficulty identifying the position of the needle tip by manual labelling of the US images. The distance between the two positions is approximately 3.5 mm.

**Figure 13 sensors-22-09035-f013:**
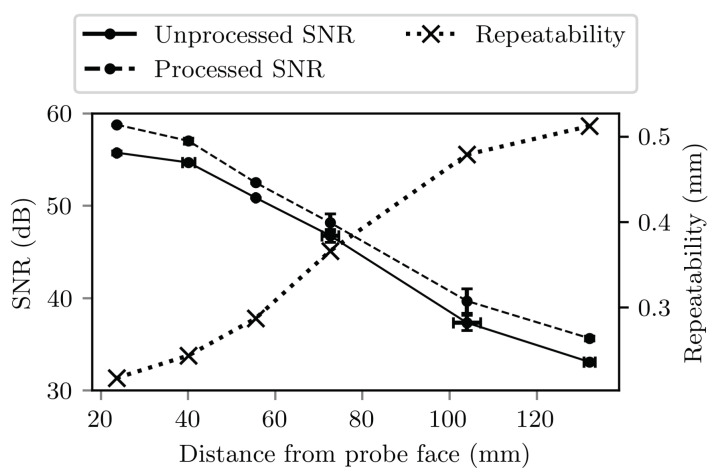
Signal-to-noise ratio and tracking repeatability magnitude measured at six depths in the bovine tissue phantom. Depths were determined using the UNT system. Vertical error bars show standard deviations of three repeat measurements of signal-to-noise ratio. Horizontal error bars on the unprocessed signal-to-noise ratios shown standard deviations in the depth measurement across the three repeats, and are applicable to all three plotted curves.

**Figure 14 sensors-22-09035-f014:**
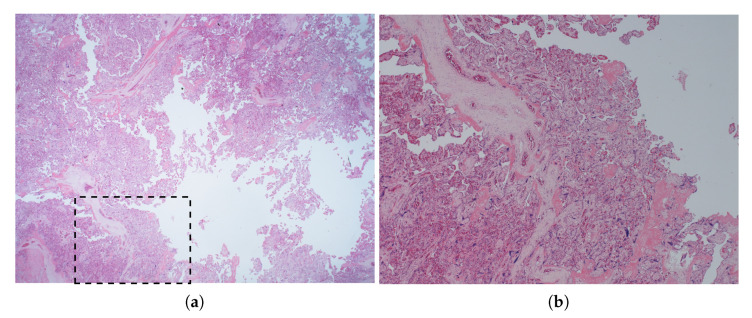
×12 (**a**) and ×40 (**b**) magnification photomicrographs of a representative histological section of placenta after needle insertion and exposure to light from a bare fibre, demonstrating physical villus separation at the needle track but no heat-related tissue damage. The zoomed section shown in (**b**) is indicated by the dashed rectangle in (**a**).

**Table 1 sensors-22-09035-t001:** Magnitude and directional components of the spatial-mean and spatial-standard-deviation of tracking error and repeatability measured in the homogenous medium (water). The tangential direction is orthogonal to the radial direction.

	Error Vector	Absolute Error	Repeatability
Lateral	−0.34 ± 0.58 mm	0.50 ± 0.45 mm	0.26 ± 0.15 mm
Axial	0.08 ± 0.41 mm	0.34 ± 0.24 mm	0.09 ± 0.07 mm
Radial	−0.02 ± 0.38 mm	0.32 ± 0.21 mm	0.04 ± 0.02 mm
Angular	−0.13 ± 0.22${}^{{}^{\circ}}$	0.19 ± 0.16${}^{{}^{\circ}}$	0.10 ± 0.05${}^{{}^{\circ}}$
Tangential	−0.34 ± 0.57 mm	0.51 ± 0.46 mm	0.27 ± 0.17 mm
Magnitude	0.67 ± 0.41 mm	—	0.28 ± 0.17 mm

## Data Availability

The data that support the findings of this study are available from the corresponding authors upon request.

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
