# Peer review of "Intraoperative Needle Tip Tracking with an Integrated Fibre-Optic Ultrasound Sensor"

_sensors, 2022, doi:10.3390/s22239035_

Round 1
Reviewer 1 Report
I thinks this paper is of sure interest for your journal.
I find the paper well written. The title is clear and the explanations exhaustive.
The authors described the topic in an appropriate and clear way.
I believe that this paper can surely be considered suitable for publication without further revisions.
Author Response
Many thanks for your review.
Reviewer 2 Report
The paper “Intraoperative Needle Tip Tracking with an Integrated Fibre-optic Ultrasound Sensor” is devoted to a new precise method of tracking the position of a needle tip under US control during medical procedures like tumor biopsy. For precise tracking, the authors use a bespoke Fabri-Perot US sensor at the needle tip. The idea of using a Fabri-Perot sensor for needle track has been previously reported by the authors of this paper, and the current study is the continuation of their previous findings. The authors report on the results of a comprehensive study on the accuracy of assessing the coordinates of the needle tip on US B-scans. The developed approach shows the high accuracy of needle tip tracking – the tracking error is about 1 mm, which is less than a needle diameter, and the tracking stability is very high for different insertion angles and needle tip positions against the US probe position. The suggested approach can potentially improve needle navigation during a biopsy procedure using a standard US imaging device and probe.
The paper is well-written and scientifically sound, however, some points need consideration before accepting the paper.
1. What about the precision of the applied UNT procedure if applied with other US devices and probes? Does it need additional calibration procedures? A displacement of 3.5mm between the true and assessed needle tip position in the direction of the needle shaft was found. This displacement might be different for different US probes and devices. The scanning geometry is also can be different. So, the universality of the developed approach may not be obvious.
2. As I understand, the main benefit of the Fabri-Perot sensor is its wide band properties in comparison to piezoelectric transducers, and there is no need to make different transducers for different US bands used for US imaging. However, this specific procedure uses not many of them, and 2-3 transducers will cover all possible frequency bands (please, correct the number if I mistake). In addition to this, piezoelectric transducers are used in different invasive medical procedures and the FDA process would pass easier in comparison to new specific materials applied in this study. Finally, the Fabry-Perot needs an expensive SS laser that could make the procedure more expensive.
3. The parts devoted to laser heating investigation in the case of Fabry-Perot tip damage look strange. First, why tests were made with the laser power of 25 mW, while UNT uses only ~3 mW, which is low enough to fit laser safety requirements? Second, the statement “Laser-induced heating may be higher in tissue than in water, where there is no convection acting to remove heat from the location of the source” looks strange because heat is carried away with the bloodstream quite effectively in the human body. The penetration depth of 1500-1600 radiation is large enough to reach blood vessels with sufficient blood velocity. Third, I suppose that if the Fabry-Perot tip is damaged and some piece of it penetrates the human body, it will definitely be more dangerous than tissue heating. In this regard, it would beneficial also to discuss the safety of the Fabry-Perot tip, its possibility for sterilization (or a single-use is supposed), and all other requirements for medical supplies, involved in invasion medical procedure.
Minor
1. Needle tip and needle mount are hard to be seen in Fig. 4
2. Figure 8 caption: exclude one “measured”
3. Please, indicate what template (delayed) signal you use for the match filter. Is it the signal received from the US system corresponding to the voltage waveform applying to a US probe?
4. It is difficult to interpret the location of the needle shaft in Fig. 12 (c). So, please, indicate it by a thin/dashed line. Also, please, indicate the insertion angle and scale bar. The authors mentioned that a needle tip is hard to be visualized on US images for steep insertion angles, but it seems that the insertion angle in this Figure is near 0.
Author Response
Many thanks for your thorough review. Please see attachment for responses.

Reviewer 3 Report
The article intraoperative needle tip tracking with an integrated fiber-optic ultrasound sensor deals with advancing a marker system to highlight the needle tip during surgery. This work reports the next step of the work proposed by the same group.
As the authors state, the differences in sound speed affect the image and tracked position. Therefore, is there a physical limit that affects the proposed technique? Could this physical limit avoid further enhancement in the proposed technique? Perhaps you can answer this question in the discussion section.
Can the authors state the maximum permissible latency explicitly? to compare how the results of this work can be affected by such limitations?
If possible, add labels of lateral and axial (x,y) directions in figure 4.
The number of references is adequate; many are from the last five years. No excessive self-citation was found.
Author Response
Many thanks for your review. Please find each of your comments addressed below.
As the authors state, the differences in sound speed affect the image and tracked position. Therefore, is there a physical limit that affects the proposed technique? Could this physical limit avoid further enhancement in the proposed technique? Perhaps you can answer this question in the discussion section.
As explained on line 647 (Discussion) of the previously submitted version of the manuscript, the aim of the tracking system is to locate the position the needle tip within the US image. To do this, the tracking system must make the same incorrect assumption of homogenous sound speed that the imaging system does. Any difference between the assumed and true sound speed affects the tracking system and imaging system equally, and therefore there is no physical limit of sound speed error.
Can the authors state the maximum permissible latency explicitly? to compare how the results of this work can be affected by such limitations?
We don’t have any information on the maximum permissible latency for such a procedure. The needle is typically inserted quite slowly towards the target, and from our experience working with clinicians on the project we understand that while it is noticeable when the needle changes direction, it is not currently an issue. The penultimate paragraph of the discussion section states that latency can be reduced to near-imperceptible levels through the use of an alternative frame grabber.
If possible, add labels of lateral and axial (x,y) directions in figure 4.
Spatial axes have been added to figure 4 (and the image has been changed, as requested by another reviewer).